# No Evidence for Myocarditis or Other Organ Affection by Induction of an Immune Response against Critical SARS-CoV-2 Protein Epitopes in a Mouse Model Susceptible for Autoimmunity

**DOI:** 10.3390/ijms24129873

**Published:** 2023-06-08

**Authors:** Rebecca Maria Ignatz, Vanessa Antje Zirkenbach, Mansur Kaya, Vera Stroikova, Renate Öttl, Norbert Frey, Ziya Kaya

**Affiliations:** 1Department of Cardiology, University of Heidelberg, 69120 Heidelberg, Germany; rebecca.ignatz@med.uni-heidelberg.de (R.M.I.); vanessa.zirkenbach@med.uni-heidelberg.de (V.A.Z.); mansur.kaya@med.uni-heidelberg.de (M.K.); vera.stroikova@med.uni-heidelberg.de (V.S.); renate.oettl@med.uni-heidelberg.de (R.Ö.); norbert.frey@med.uni-heidelberg.de (N.F.); 2DZHK (German Centre for Cardiovascular Research), Partner Site Heidelberg/Mannheim, University of Heidelberg, 69120 Heidelberg, Germany

**Keywords:** SARS-CoV-2, long-COVID, myocarditis, cardiovascular damage, autoimmune

## Abstract

After *Severe acute respiratory syndrome coronavirus 2* (SARS-CoV-2) developed into a global pandemic, not only the infection itself but also several immune-mediated side effects led to additional consequences. Immune reactions such as epitope spreading and cross-reactivity may also play a role in the development of long-COVID, although the exact pathomechanisms have not yet been elucidated. Infection with SARS-CoV-2 can not only cause direct damage to the lungs but can also lead to secondary indirect organ damage (e.g., myocardial involvement), which is often associated with high mortality. To investigate whether an immune reaction against the viral peptides can lead to organ affection, a mouse strain known to be susceptible to the development of autoimmune diseases, such as experimental autoimmune myocarditis (EAM), was used. First, the mice were immunized with single or pooled peptide sequences of the virus’s spike (SP), membrane (MP), nucleocapsid (NP), and envelope protein (EP), then the heart and other organs such as the liver, kidney, lung, intestine, and muscle were examined for signs of inflammation or other damage. Our results showed no significant inflammation or signs of pathology in any of these organs as a result of the immunization with these different viral protein sequences. In summary, immunization with different SARS-CoV-2 spike-, membrane-, nucleocapsid-, and envelope-protein peptides does not significantly affect the heart or other organ systems adversely, even when using a highly susceptible mouse strain for experimental autoimmune diseases. This suggests that inducing an immune reaction against these peptides of the SARS-CoV-2 virus alone is not sufficient to cause inflammation and/or dysfunction of the myocardium or other studied organs.

## 1. Introduction

SARS-CoV-2 started its global journey in Wuhan, China, over two years ago. In December 2019, a new respiratory disease spreading over national borders was reported [1,2]. Initial investigations described it as a new member of the *Coronaviridae* family sharing a high sequence identity with SARS-CoV, a pathogen involved in an outbreak years ago [3]. Because of its close degree of relationship to other coronaviruses, SARS-CoV-2 could have possible effects on the heart as it was reported for the *Middle east respiratory syndrome coronavirus* (MERS-CoV), which was associated with acute myocarditis and heart failure [4]. Indeed, several groups reported the presence of viral particles in the myocardium and involvement of the heart during a SARS-CoV-2 infection, including myocardial inflammation with an increase in high-sensitive troponin T (hsTnT) levels and C-reactive protein (CPR) [5,6,7,8,9,10]. These findings show a potential for the heart to be directly affected. SARS-CoV-2 infection can also lead to inflammation of the endothelium, which serves as a protective layer in blood vessels. This can lead not only to pneumonia but also to systemic endothelitis, which subsequently spreads to the heart [11,12,13]. This cardiac injury caused by different mechanisms is favored by the fact that the virus enters the cells via three different pathways, including angiotensin-converting enzyme 2 (ACE-2) and neutropolin-1 (NRP-1) receptor or vesicle-mediated. SARS-CoV-2 is mainly taken up via the ACE-2 receptor, which can be found in the respiratory system, gastrointestinal tract, kidney, and liver, but also in the heart muscle [14,15,16,17]. In addition to that, NRP-1 can serve as another entrance into the first layer of the epithelium. This receptor, responsible for the regulation of cardiovascular, neuronal, and immune response processes [18,19], is interfered with by the binding of SARS-CoV-2. Blocking this interaction using RNA or monoclonal antibodies reduces SARS-CoV-2 infectivity [20,21,22]. Another mechanism for virus uptake involves vesicle-mediated endocytosis, which is similar to the uptake via the ACE-2 receptor and has been demonstrated in in vivo experiments [21,22]. Thereby, the surface proteins of the virus play an important role by harboring binding motifs for NRP-1 and ACE-2 receptors [21,23].

With regard to the progression of the infection, the majority of infected individuals experience milder symptoms with no necessity for hospitalization. However, both hospitalized (50–70%) and not hospitalized (10–30%), as well as a small group of vaccinated (10–12%) individuals, have a certain risk for developing nonspecific persistent symptoms such as fatigue and dyspnea, as well as impairment of several organ systems [24,25,26,27,28,29] depending on the patient’s age and the severity of the initial SARS-CoV-2 disease [30,31,32,33,34]. This disease sequel is colloquially termed “long-COVID” and is defined by the Centers for Disease Control and Prevention (CDC) as a collection of symptoms developed during or following a SARS-CoV-2 infection [35]. About the severity and variety of symptoms, little is known, and no one can predict how long it will take until normal health status is reached again [36]. A common long-COVID effect involves the cardiovascular system, which among others, appears as a stroke, inflammatory heart disease, and others [26,29,37,38,39,40]. An analysis by the US Department of Veterans Affairs databases includes more than 150,000 individuals and indicates an increased risk for heart failure, dysrhythmias, and stroke [37]. Additionally, cardiac involvement and abnormalities were found in 78% of 100 individuals after a worse SARS-CoV-2 infection and in 58% of participants with long-COVID [5,41]. Not only the heart but other organ systems can be affected by long-COVID, even in low-risk individuals. One study showed damage to at least one of the tested organs (heart, lungs, liver, kidneys, pancreas, and spleen) in 70% of 201 patients and multiorgan damage in 29% of subjects [42]. A subsequent study with 536 participants found single-organ damage in 59% and multiorgan damage in 27% [43]. Further, a large-scale kidney study showed an increased risk of adverse kidney outcomes after SARS-CoV-2 infection [44]. As potential risk factors for the development of long-COVID, female sex, diabetes, viral interactions and pre-exposure to a previous SARS-CoV-2 infection, and the presence of specific autoantibodies against, e.g., G-protein coupled receptors (ACE-2) or interferons, were identified [29,45,46,47,48]. Nevertheless, the reasons are diverse or probably overlapping. These include, for example, the development of viral reservoirs in tissues [49,50], alteration of the microbiome or the virome [49,51,52], and microvascular blood clotting with endothelial dysfunction [49,53,54,55], but also modulation of autoimmunity [45,48,49,56] and priming of the immune system by molecular mimicry [49]. Thereby, Murphy et al. suggested persisting antibodies followed by the production of anti-idiotype antibodies as a cause for long-COVID [57]. These antibodies are able to bind idiotype sites of antibodies and share a similarity with the initial antigen, which in turn could potentially lead to the induction of autoimmune responses against the body’s own organs, including the myocardium, via cross-reactivity or epitope-spreading [58,59]. In the case of SARS-CoV-2 infection, antibodies against the spike protein, which is able to modulate ACE2 signaling, are produced. If idiotype antibodies, which in a SARS-CoV-2 infection resemble the spike protein, are formed, they could stimulate the ACE2 receptor like the original antigen and contribute to the development of long-COVID [57]. The way antibodies are formed and the final titer after SARS-CoV-2 infection and vaccination also depend on the sex, each of which differs in its seroreversion. Here, women show a lower total titer and a decreasing antibody level compared to men [29,60,61]. 

Although there are several diagnostic approaches, they are not very specific because the underlying mechanisms of long-COVID are not yet well understood. Thus, it is important to understand the pathomechanism in order to develop possible therapeutic approaches and to improve diagnostics. Therefore, we wanted to address the hypothetical question of whether the induction of an immune response against one or a pool of these SARS-CoV-2-virus spike-, membrane-, nucleocapsid-, or envelope-peptide sequences could induce an immune response against the myocardium and/or other organs leading to inflammation or dysfunction of these organs by cross-reactivity or epitope spreading. Which sequences lead to the formation of antibodies can be found using sequence analyses or comparisons with related species. Based on their amino acid sequence homology, Grifoni et al. predicted potential immune-reactive epitopes for B-cells by the comparison of SARS-CoV-2 with SARS-CoV. Both viruses share high similarity in phylogenetic terms and in the level of sequence identity. They found that certain SARS-CoV regions were dominant for B-cell responses and that those regions were well conserved in terms of sequence with SARS-CoV-2 [62]. These regions are primarily located in the spike protein but also in the membrane, nucleocapsid, and envelope protein. The spike protein mediates the uptake of SARS-CoV-2 and is, therefore, in direct contact with the host system [63,64]. For the virus assembly, the membrane protein is responsible, which interacts with nucleocapsid protein and RNA [65]. The latter are held together and organized by the nucleocapsid protein as the viral genome [66]. Therefore, we used the predicted B-cell reaction dominant sequences to produce peptides which were then used for immunization of a mouse strain known to be very susceptible to the induction of experimental autoimmune diseases such as EAM [67,68] in order to investigate immunologic side effects and long-term damage of the heart and other organs after contact with immunorelevant sequences of SARS-CoV-2. 

## 2. Results

### 2.1. Effects of the Immune Reaction against the Single B-Cell Reaction Dominant SARS-CoV-2 Spike Protein Peptides and Peptide Pools on the Myocardium

To study the effect of the immune reaction induced with immunization with the B-cell response dominant SARS-CoV-2 spike protein sequences on the heart, 5-week-old A/JOla mice, which are very susceptible to the development of autoimmune diseases, were immunized with 150 µg peptide emulsion (n = 6–7, Appendix A) once a week for three times, as described before [67,69,70]. For control, Troponin I (TnI) peptide-treated mice (n = 10) for positive peptide-induced myocardial inflammation and untreated mice (n = 4) as negative control were used. To boost the immune response with the spatial arrangement of the SP sequences, which might have a higher immunogenic and, thus, inflammatory effect on the myocardium, we combined consecutive single SARS-CoV-2 sequences of spike protein into clusters (SPG1–SPG5). These peptide pools were administered to A/JOla mice equally used as in the experiments before (SPG n = 6, TnI peptide n = 7, untreated n = 6, Appendix A).

As a direct demonstration of inflammation, we investigated the infiltration of immune cells into the heart tissue with hematoxylin and eosin (HE). All analyzed tissue sections of SP-treated mice showed no in-filtrating cells compared to control mice, whereas positive control TnI mice showed a significant inflammation in the myocardium (11.62 ± 3.757, * *p* < 0.05; Figure 1A, left). As seen in the single peptide experiment, a microscopic evaluation of the heart sections showed no inflammation in the myocardium of animals immunized with pooled SP peptides (Appendix A), while TnI-peptide-treated mice developed a myocardial inflammation (49.71 ± 7.655 %, ** *p* < 0.01; Figure 1A, right). Representative images of the 2- and 20-fold magnified heart sections for both SP groups are shown in Figure 1B and Appendix A.

To investigate cardiac damage, levels of hsTnT, a cardiac protein that is released into the bloodstream after cardiac damage [71,72], were measured. Animals treated with SP sequences were negative for hsTnT (Figure 2A). In SPG1–SPG5 treated and untreated animals, hsTnT were negative as well, whereas TnI-peptide-treated mice exhibited significantly elevated hsTnT markers (356.7 ± 153.7 pg/mL, ** *p* < 0.01, Figure 2B). Due to a detection limit of 5 pg/mL and our dilution of 1:10, values < 50 pg/mL are considered negative. Additional evaluation of cardiac dysfunction via echocardiographic examination of the ejection fraction displayed no significant decrease in the pump function in neither pooled SP peptide group during the 28-day period (Figure 2C). Because immunization with viral proteins induces the production of antibodies, we checked the antibody titer against the pooled SP peptides (SPG) used for immunization. In the serum of the SP-treated mice, the production of antibodies against the respective SP cluster (SPG1–SPG5) was detected in the pooled SP-treated mice with the highest titer in SPG1 (1:683,467 ± 101,151, *p* < 0.0001; Figure 2D), which consists of the sequences SP01–SP06 localized in the N-terminal region (Appendix A). 

### 2.2. Effects of the Immune Reaction against the Single B-Cell Reaction Dominant SARS-CoV-2 Membrane Protein Peptides and Peptide Pools on the Myocardium

Next, we studied the effect of SARS-CoV-2 membrane protein sequences on the heart; 5-week-old A/JOla mice were immunized with 150 µg peptide emulsion (n = 6) once a week for three times. For control, TnI-peptide-treated mice (n = 10) for positive and untreated mice (n = 4) as negative control were used. To boost the immune response here as well, we clustered single MP sequences together for membrane protein pool (MPG), which were administered to A/JOla mice (n = 6) equally used as in the experiments before. We found no inflammation in the heart tissue sections of MP immunized and untreated mice, while our positive TnI peptide control mice (Figure 3A, left). The microscopic evaluation of the heart showed no signs of inflammation except for the animals treated with TnI peptide (49.71 ± 7.655 %, *p* < 0.05; Figure 3A, right). Representative images of MP and MPG-treated hearts are shown in Figure 3B and Appendix A.

The investigation of cardiac damage in MP-treated mice showed no detectable serum hsTnT (Figure 4A, left). Further, the hsTnT levels of MPG-immunized mice were negative, while TnI-peptide-positive controls showed elevation in this marker (Figure 4A, right). The echocardiographic examination of the ejection fraction for cardiac dysfunction assessment showed no restriction of the pump function in MPG-treated mice over a 28-day period (Figure 4B). Equal to the spike protein, the immunization with pooled MP led to the production of antibodies against the respective protein sequence. The MPG IgG antibody titer in MPG-immunized mice turned out lower compared to the SPG groups (MGP 1:1333 ± 608.1, SPG1–4: ** *p* = 0.0022, SPG5: * *p* = 0.013; Figure 2C and Figure 4C). 

### 2.3. Effects of the Immune Reaction against the Single B-Cell Reaction Dominant SARS-CoV-2 Nucleocapsid Protein Peptides and Peptide Pools on the Myocardium

Next, we studied the effect of SARS-CoV-2 nucleocapsid protein sequences on the heart; 5-week-old A/JOla mice were immunized with a 150 µg peptide emulsion (n = 6) once a week for three times. For control, TnI-peptide-treated mice (n = 10) for positive and untreated mice (n = 4) as a negative control were used. To boost the immune response here as well, we clustered single NP sequences together for nucleocapsid protein pool (NPG), which were administered to A/JOla mice (n = 6) equally used as in the experiments before.

Immunization with NP peptides did not lead to the development of inflammation in the myocardium, while the TnI-immunized mice showed infiltration of immune cells into the myocardium (Figure 5A, left). Similar results were achieved in NPG-immunized mice (TnI 49.71 ± 7.655 %, ** *p* < 0.01; Figure 5A, right). Representative images of NP and pooled NP immunized heart sections are shown in Figure 5B and Appendix A.

No hsTnT levels could be detected in the serum of animals immunized with NP (Figure 6A, left). Compared to the hsTnT levels of the untreated animals or those treated with the pooled viral peptides, the hsTnT levels of control mice treated with TnI peptide were significantly elevated (356.7 ± 15.7 pg/mL, ** *p* < 0.01, Figure 6A, right). The analysis of cardiac dysfunction via echocardiographic examination of the ejection fraction from animals immunized with the pooled viral peptide sequences was assessed over a 28-day period and showed no effect on the pump function (Figure 6B). The animals treated with NPG showed a similarly high antibody formation as the animals that received the SPG peptides (NPG 1:14,000 ± 7734; Figure 2C and Figure 6C). 

### 2.4. Effects of the Immune Reaction against the Single B-Cell Reaction Dominant SARS-CoV-2 Envelope Protein Peptides and Peptide Pools on the Myocardium

Finally, we studied the effect of the SARS-CoV-2 envelope protein sequences on the heart; 5-week-old A/JOla mice were immunized with a 150 µg peptide emulsion (n = 6) once a week for three times. For control, TnI-peptide-treated mice (n = 10) for positive and untreated mice (n = 4) as a negative control were used. To boost the immune response here as well, we clustered single EP sequences together for an envelope protein pool (EPG), which was administered to A/JOla mice (n = 6) equally used as in the experiments before.

Inflammation in EP-treated mice could not be observed except for TnI-peptide-treated mice (Figure 7A, left). The microscopic evaluation of the EPG-treated heart sections showed no signs of inflammation except for the animals treated with TnI peptides (Figure 7A, right). Representative images of EP and pooled EP immunized heart sections are shown in Figure 7B and Appendix A.

Further, the hsTnT values for animals treated with EP sequences were negative, as well as in EPG-treated mice (Figure 8A). The echocardiographic examination of the ejection fraction from animals immunized with the EPG was measured and showed no significant decrease in the pump function could be observed over the 28-day period (Figure 8B). In the course of immunization with EPG sequences, autoantibodies against this EPG pool were produced with the lowest titer of all SARS-CoV-2 peptide pools (1:66.67 ± 42.16; Figure 8C).

### 2.5. Effects of the Immune Reaction against the Pooled B-Cell Reaction Dominant SARS-CoV-2 Peptides on the Liver, Kidney, Lung, Intestine, and Muscle

Because of its potential to damage multiple organs [29,42,43,44], we checked other organs (liver, kidney, intestine, lung, and muscle) after immunization with SARS-CoV-2 peptide pools of spike, nucleocapsid, membrane, and envelope proteins. For detection of inflammation, tissue sections of the organs were stained with HE, and, as far as assessable, liver, kidney, lung, intestine, and muscle tissue showed no abnormalities or inflammation compared to untreated animals (Figure 9A and Appendix A). Representative images of the liver, kidney, lung, intestine, and muscle from SPG-, NPG-, MPG-, and EPG-treated mice are shown in (Figure 9C and Figure 10). In addition, albumin, urea, creatinine, creatine kinase MM (CKM) and aspartate aminotransferase (AST) levels were determined in the serum of the animals via ELISA to investigate some organ functions. In sum, all detected concentrations of serum markers were located in a normal range, and we found no pathologic plasma levels for impairment of liver, kidney, or muscle function [73,74,75,76,77,78,79]. Urea plasma levels, which normally range between 30–50 mg/dL [75,76,77], were scattered, and we could detect a significant difference between SPG4 (48.48 ± 2.251 vs. 56.07 ± 2.056 (control), * *p* = 0.0341, *t*-test) and MPG (46.52 ± 3.339 vs. 56.07 ± 2.056 (control), * *p* = 0.0209, *t*-test; Figure 9B) compared to untreated control. Albumin levels in corona-peptide-pool-immunized mice were lower compared to untreated controls. Only the NPG group significantly reduced (3312 ± 198 vs. 5058 ± 634.8; * *p* = 0.0485, *t*-test; Figure 9A). A normal albumin level is between 20–50 g/L [73,74]. Our results showed increased creatinine concentrations in SPG1 and SPG3 mice compared to untreated control (SPG1: * *p* = 0.0172, SPG3: * *p* = 0.0369, *t*-test). All values were close to or below the lower limit of detection (Figure 9B). Normally, creatinine lies around 1 mg/dL [75]. AST levels in corona-peptide-pool-immunized mice did not differ significantly from untreated control animals, except for the group treated with SPG1, which showed slightly lower AST levels (SPG1 1.75 ± 0.2277, control 2.718 ± 0.3009, * *p* = 0.0458, *t*-test; Figure 9B). We were not able to find a measurable plasma level CKM (Appendix A).

## 3. Discussion

After SARS-CoV-2 virus infections became a global pandemic in 2019, whose consequences are still affecting people’s health conditions, the need for research on potential vaccines and treatments increased [80,81,82]. An important issue is the exploration of the underlying mechanisms to understand, prevent, or treat the increasing number of side effects of the infection [83,84,85]. One of these complications is the impact of the virus on the heart, not only in older people with relevant pre-existing conditions but also in younger patients. Increasingly, heart damage and myocarditis have occurred as a result of a SARS-CoV-2 infection [86,87]. The surface proteins the virus uses to enter cells via the ACE-2 or NRP-1 receptor can also serve as recognition sequences for the host’s immune system. As a result, these sequences can also play a key role in the path of the disease [88]. 

Not only acute infection can cause damage to the affected organ systems, but also long-term damage can occur after exposure to SARS-CoV-2 (long-COVID). The exact pathomechanisms have not yet been fully understood. Therefore, our group investigated the effects of the relevant peptide sequences of the virus on the development of myocarditis in a mouse strain (A/JOla) very susceptible to autoimmune diseases such as EAM [89]. In particular, it should be investigated whether the immune response to the immunogenic viral peptide sequences can lead to organ damage, inflammation, or dysfunction. Therefore, we first administrated immunogenic SARS-CoV-2 peptides from spike, membrane, nucleocapsid, and envelope protein to A/JOla mice, susceptible to autoimmune diseases, in order to investigate their pathogenic immunologic potential on the heart and other organs (lung, liver, kidney, intestine, and muscle). Neither histological examination of the hearts nor markers such as hsTnT in the serum nor echocardiographic examination of the animals showed signs of inflammation, heart damage, or heart dysfunction as a consequence of an immune response to viral spike, membrane, nucleocapsid, and envelope protein peptides. Since there are negative hsTnT levels and no signs of inflammation of the myocardium or other organs in mice treated with the viral single peptides, treatment with B-cell reaction-relevant SARS-CoV-2 peptide sequences does not appear to have a pathogenic immunological effect.

To boost the immune response, we immunized mice with pooled peptide sequences. Again, no signs of inflammation were found in the myocardium. As reported by Raman et al. and Hasan et al., other organs can also be damaged as a result of an acute SARS-CoV-2 infection [85,90]. In addition, a common long-COVID effect involves the cardiovascular system, which among others, appears as a stroke, inflammatory heart disease, and other diseases [26,29,37,38,39,40]. Not only the heart but other organ systems can be affected by long-COVID. Studies showed single (70%) and multi-organ damage (29%) in either/or heart, lung, liver, kidney, pancreas, or spleen of the examined patients, as well as increased risk for adverse kidney outcomes after SARS-CoV-2 infection [43,44]. Thus, the liver, kidney, lungs, intestines, and muscles were examined via histopathology. However, no inflammation was detected as a result of immunization with the pooled viral peptide sequences. In the course of that, we additionally checked whether there were serum markers altered that refer to organ function. Albumin in humans is the most abundant protein circulating in the blood and operates as a regulator of the hydrologic balance and small molecule transport. It is produced in the liver and serves as a marker for liver synthesis [91,92]. Here we could not detect any damage as the values lay in a normal range [73,74]. For kidney function, urea is a common marker. As a waste product in protein digestion, urea can indicate kidney malfunction [93,94]. In line with the histopathological examination, we found no abnormalities in urea plasma concentration compared to the reported values [75,76,77]. As a waste product of creatine degradation, creatinine is a marker for kidney function as it is cleared via glomeruli filtration [94,95,96]. Our determined values were close to or below the detection limit and are, therefore, inconclusive and evaluable [77]. In comparison to the heart muscle, the integrity of skeletal muscle was also assessed. Therefore, CKM was used as a marker. Creatine kinase is responsible for cellular energy metabolism by providing ATP and, with this, chemical energy for muscle power [78,79]. We also measured AST in the serum of the mice. AST is an enzyme found mainly in the liver, where it catalyzes the conversion of α-ketoglutarate to the amino acid glutamic acid [97]. The detection of AST in the blood serves as a marker for liver damage [98,99]. It has also been found to be elevated in serum following injuries of skeletal muscle as well as after cardiac damage [100,101]. Our measured AST levels did not indicate either liver damage or signs of myocardial or skeletal muscle injury as a result of immunization with the SARS-CoV-2 peptides, as they were not increased compared with the untreated control group. In conclusion, immunization with SARS-CoV-2 pooled peptide sequences does not lead to damage in various organs. This leads to the assumption that there are additional risk factors that support organ damage, including cardiac dysfunction or inflammation following a SARS-CoV-2 infection. In addition, differences were found regarding sex, with males at the median age of 54.8 years being affected more frequently [102]. Regarding potential risk factors for the development of long-COVID, female sex, diabetes, viral interactions and pre-exposure, and the presence of specific autoantibodies were identified [29,45,46]. 

As far as the limitation of our work is concerned, we are dealing with a special case that investigates the immune response to viral peptides as a cause of pathogenic effects on the heart and other organs. While this provides insight into the role of this particular pathomechanism, it cannot be extrapolated one-to-one to humans, as mice serve as a model organism and the peptide concentration to which a human is exposed during SARS-CoV-2 infection is very individual. However, our work provides a good comparison between an immune response to the viral SARS-CoV-2 peptides to an immune response to a cardiac TnI peptide in a well-established animal model for myocarditis. Nevertheless, the experiments depend on the resulting protein depot, which continuously releases peptides to the organism throughout the experimental period. Since, during an infection with SARS-CoV-2, the organism is confronted with the total SARS-CoV-2 proteins, the predicted epitopes could also be too short or require an even larger spatial arrangement to show an effect on the organism. From our preliminary work with the TnI model in autoimmune myocarditis, we know that there is some failure rate in developing inflammation. Therefore, there is a residual risk that our number of animals used was too small.

In conclusion, inducing an immune response by immunization of mice with different immunogenic sequences/epitopes of the SARS-CoV-2 virus spike, membrane, nucleocapsid, or envelope protein did not induce inflammation or dysfunction in the myocardium or other organs like the liver, kidney, lung, intestine, and muscle. This provides evidence that cardiac or organ damage resulting from a SARS-CoV-2 infection is not entirely caused by a reaction due to the protein sequences of the virus. This represents an important finding about the underlying mechanisms of the dangerous side effects and long-time consequences of SARS-CoV-2 infection. These findings could be important not only with regard to SARS-CoV-2 infection and the development of long-COVID symptoms but also to other viral diseases and their treatment. Further work should investigate which risk factors promote the development of myocardial damage or harming of other organ systems resulting from SARS-CoV-2 infection and the underlying molecular mechanisms.

## 4. Materials and Methods

### 4.1. Peptide Establishment

On the basis of the publication of Grifoni et al. [62], potential B-cell epitopes were generated from coronavirus spike, membrane, and envelope proteins. The predicted sequences were mapped on the whole spike, membrane, or envelope protein. The sequences were re-searched from the NCBI database (https://www.ncbi.nlm.nih.gov/protein/BCA87361.1, accessed on 4 June 2023). Due to the form of administration and stability, shorter sequences were extended with a few amino acids at the tails based on the whole sequence, and larger sequences were divided into overlapping shorter parts. These final 38 peptides were synthesized by Peptide Specialty Laboratories GmbH (Heidelberg, Germany). To investigate the effect of pooled sequences, single sequences were combined, leading to a pool of spike protein (SPG1-5), nucleocapsid protein (NPG), membrane protein (MPG), and envelope protein (EPG, Appendix A).

### 4.2. Animals and Immunization with Peptides

Female (n = 6–7) and male (n = 6) A/JOla wild-type mice, a susceptible mouse strain for autoimmune diseases, were obtained from Envigo (Huntingdon, Cambridgeshire, UK) and were maintained in the animal facility unit of the University of Heidelberg. In all experiments, 4–7-week-old mice were used. The animal study was reviewed and approved by the Institutional Review Board/ Ethics Committee of the regional council of Karlsruhe (Animal application G-4/21; date of approval 24 March 2021) and by the Animal Care and Use Committee of the University of Heidelberg (German Animal Protein Code). Based on the Directive of the European Parliament and the Council for the Protection of Animals used for scientific purposes, this study was conducted in accordance with the German animal welfare act (Directive 2010/63/EU).

To immunize female mice with single SARS-CoV-2 peptides, a 150 µg generated peptide (SP1-38, Appendix A) was used. For the immunization of male animals with pooled peptides (SPG1-5, NPG, MPG, EPG; each n = 6), 150 µg of every single peptide in supplemented complete Freud’s adjuvant with 5 mg/mL *Mycobacterium tuberculosis* H37Ra (Sigma, St Louis, MO, USA) was used. To compare the results of SARS-CoV-2 peptide immunization with the effects of myocarditis induced with cardiac TnI peptide immunization, male (n = 7) and female (n = 10) A/JOla mice at the age of 4–7 weeks were subcutaneously immunized three times at 7-day intervals (day 0, 7, 14) with 150 µg murine cardiac troponin I peptide (Peptide Specialty Laboratories GmbH) in supplemented complete Freud’s adjuvant with 5 mg/mL *Mycobacterium tuberculosis* H37Ra (Sigma), as established before in our experimental autoimmune myocarditis (EAM) mouse model [67,69,70]. As a healthy control, untreated (single peptide part n = 4, pooled peptide part n = 6) mice were used. The mice were sacrificed on day 28 using cervical dislocation. An overview of the temporal course of the experiment can be found in Appendix A.

### 4.3. Determination hsTnT

To determine the cardiomyocyte damage, serum samples were collected on day 28 when the mice were sacrificed, and hsTnT was analyzed using electrochemiluminescence (Elecsys 2010 analyzer, Roche Diagnostics, Rotkreuz, Switzerland) for myocardial damage. The assay is based on the binding of cTnT-specific monoclonal antibodies. The detection is then realized based on an electrochemiluminescence immunoassay (ECLIA) using a Tris(bipyridyl)-ruthenium(II) complex labeled as described before [103,104]. Therefore, the serum was diluted (1:10) with cold 0.9% NaCl solution. Details of the test principle have been described before [105]. This method has a blank limit of 3 pg/mL and a detection limit of 5 pg/mL [106]. Due to our dilution, values > 50 pg/mL are pathological.

### 4.4. Histopathological Analysis

The hearts of sacrificed mice were cut longitudinally vertical to the septum into two pieces, and 0.5–1 cm^2^ pieces of other organs were sampled. One-half of the heart and the organ tissues were fixed in formalin (10%) and subsequently embedded in paraffin. Sections of 2–3 µm thickness were cut and stained with hematoxylin and eosin (HE) to determine the level of inflammation using standard staining protocols and reagents. Twelve sections of each heart or six sections of the organs were inspected in a double-blinded manner by two independent investigators using light microscopy. Therefore, the area infiltrated by immune cells was considered in relation to the whole heart section. The ratio of the inflamed area to the whole section was specified in a percentage. The mean infiltration was calculated in percentage from the values of both investigators. Because the inflammation caused by immunization with TnI is cardiac-specific, it cannot serve as a positive control for other organs or tissues. Therefore, organ tissues were compared with the untreated control. If no pathological abnormalities were seen between the organ samples, a degree of inflammation of 0 was assumed. The scans of the stained organ sections were performed with a microscopic magnification of 40 times by the tissue bank of the National Center for Tumor Diseases (NCT) Heidelberg, Germany.

### 4.5. Echocardiography

Echocardiographic measurements of peptide pool immunized mice were performed using Visual Sonics Vevo 2100 system, 30 MHz linear Micro Scan transducer (MSH400), specially optimized for cardiovascular experiments in mice. Parasternal long-axis projection cine loops were acquired at the level of clear visible walls of the aortic annulus. LVEF and stroke volume were determined with appropriate software provided with the Vevo2100 platform.

### 4.6. Detection of Antibodies against Spike Protein Peptide Sequences via Enzyme-Linked Immunosorbent Assay (ELISA)

As described before, antibody titers of collected blood serum from each mouse were determined by using the ELISA technique [107,108]. Thereforse, 96-well microtiter plates (Thermo Fisher Scientific Inc., Waltham, MA, USA) were coated with 100 µL/well-pooled spike peptide sequences in bicarbonate buffer (pH 9.6) and incubated overnight. Mouse secondary IgG antibodies were diluted to 1:5000 and used for detection. Serum samples were diluted to 1:50, 1:200, 1:800, and 1:3200. Untreated mouse serum was used as a control. The color reaction was developed with TMB (KPL) (Surmodics IVD, Inc., Eden Prairie, MN, USA) and stopped by adding 100 μL of 0.3 M H_2_SO_4_ (Honeywell Holding GmbH) to each well. Optical densities were determined at 450 nm. Antibody endpoint titer for each individual mouse was calculated as the greatest positive dilution (n) of antibody (1/n).

### 4.7. Detection of Albumin, Urea, Creatinine, Creatine Kinase M, and Aspartate Transaminase

To detect albumin, a marker for liver synthesis function, in mouse sera, a double antibody sandwich ELISA (#E-90AL, ICL) was used according to the manufacturer’s manual. Absorbance was determined at 450 nm. 

For the detection of liver damage, aspartate transaminase was measured in mouse sera via a sandwich ELISA (MEB214Mu, Cloud-clone Corp., Katy, TX; USA) according to the manufacturer’s manual. Absorbance was determined at 450 nm.

Kidney filtration function was investigated with a urea detection kit using a color reaction for blood urea nitrogen BUN (QuantiChrom Urea Assay Kits, DIUR-100, BioAssay Systems, Hayward, CA, USA) according to the manufacturer’s manual. Optical density was measured at 520 nm. Additionally, a creatinine enzymatic assay (#80350, Crystal Chem, Zaandam, Netherlands) was used according to the manufacturer’s manual. Absorbance was measured at 550 nm.

To investigate skeletal muscle damage, muscle creatine kinase was determined by sandwich enzyme immunoassay technique (ELISA Kit SEA109Mu, Cloud-Clone) according to the manufacturer’s manual. Absorbance was measured at 450 nm. 

### 4.8. Statistical Analysis

Statistical analysis of the data was performed using GraphPad Prism version 7.00 for Windows (GraphPad Software, La Jolla, CA, USA). All data are plotted as individual points. The normal distribution of the control group was tested using the D’Agostino–Pearson normality test. Data summaries are given as mean ± SEM. All tests used were 2-tailed. Each SARS-CoV-2 peptide or peptide pool treated group was analyzed using the Mann–Whitney test for skewed data in comparison to the negative control. For multiple group comparisons with repeated measurements, 2-way ANOVA was performed, followed by a multiple comparison test. The threshold of significance for all tests was set at 0.05.

## Figures and Tables

**Figure 1 ijms-24-09873-f001:**
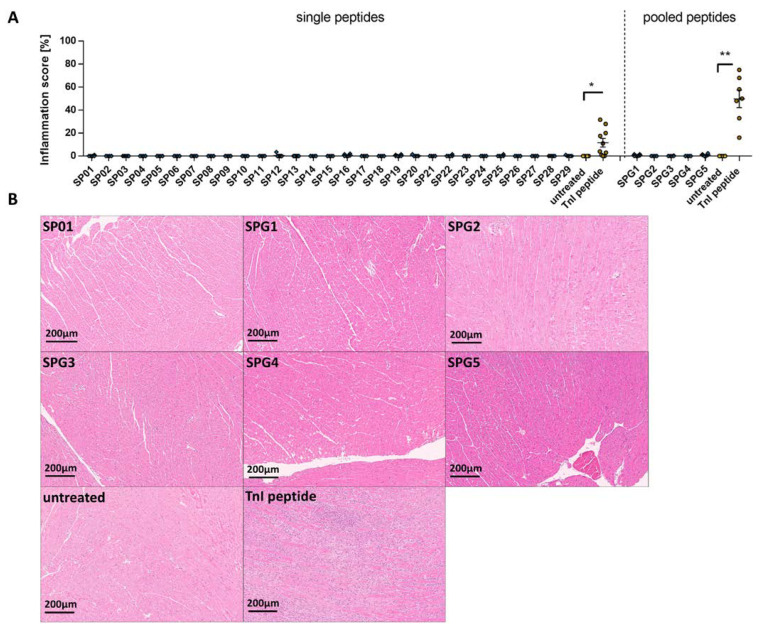
Evaluation of inflammation after immunization with SARS-CoV-2 SP peptides and peptide pools. A/JOla mice were immunized with 150 µg peptide on days 0, 7, and 14 [SP: SP peptides n = 6–7, TnI n = 10, untreated n = 4; SP pool: SPG peptide n = 6, TnI n = 7, untreated n = 6] (**A**) Inflammation score of single SP peptide (**left**) and pooled SP peptide (**right**) immunized A/JOla mice in percentage based on HE staining scored by two experienced investigators (**B**) Representative images of paraffin-embedded murine heart tissue sections of 3 µm stained with HE at a 20-fold magnification. All values are indicated as mean ± SEM. For statistics, a Mann–Whitney test was performed, and significant values were marked (* *p* < 0.05 and ** *p* < 0.01).

**Figure 2 ijms-24-09873-f002:**
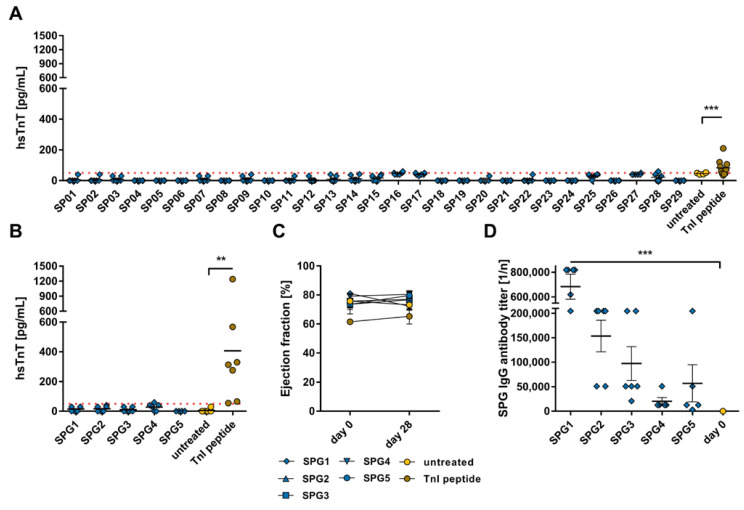
Evaluation of cardiac damage and dysfunction after immunization with SARS-CoV-2 SP peptide and SP peptide pools. A/JOla mice were immunized with 150 µg peptide on days 0, 7, and 14 [SP: SP peptides n = 6–7, TnI n = 10, untreated n = 4; SP pool: SPG peptide n = 6, TnI n = 7, untreated n = 6]. (**A**) Determination of hsTnT-levels as a cardiac damage marker in blood serum of SP and SPG peptide immunized mice collected on day 28 [threshold (dot line): 50 pg/mL]. (**B**) Determination of hsTnT-levels as a cardiac damage marker in blood serum of pooled SP peptide immunized mice collected on day 28. (**C**) Analysis of cardiac dysfunction via echocardiographic assessment of ejection fraction using M-Mode over a period of 28 days. (**D**) Antibody production is directed against SPG peptides on day 0 and after a 28-day period. All values are indicated as mean ± SEM. For statistics, a Mann–Whitney test (**A**,**B**,**D**) or ordinary two-way ANOVA (**C**) was performed, and significant values were marked (** *p* < 0.01, *** *p* < 0.001).

**Figure 3 ijms-24-09873-f003:**
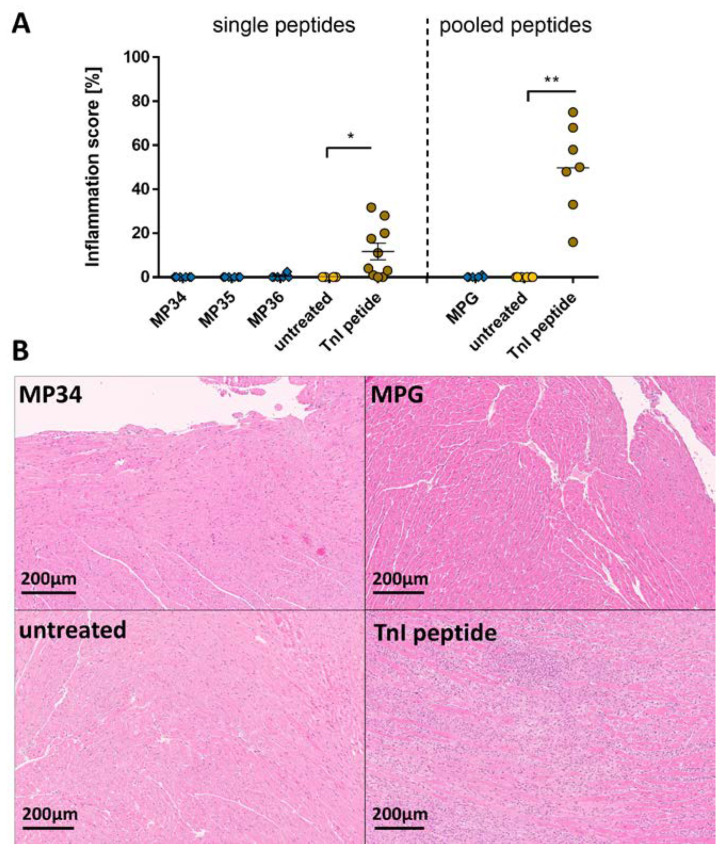
Evaluation of inflammation after immunization with SARS-CoV-2 membrane protein peptides and peptide pools. A/JOla mice were immunized with 150 µg peptide three times [MP34-36 n = 6, TnI n = 10, untreated n = 4; MPG n = 6, TnI n = 7, untreated n = 6]. (**A**) Inflammation score of single MP peptide (**left**) and pooled MP peptide (**right**) immunized A/JOla mice in percentage based on HE staining scored by two experienced investigators (**B**) Representative images of paraffin-embedded murine heart tissue sections of 3 µm stained with HE at a 20-fold magnification. All values are indicated as mean ± SEM. For statistics, a Mann–Whitney test was performed, and significant values were marked (* *p* < 0.05 and ** *p* < 0.01).

**Figure 4 ijms-24-09873-f004:**
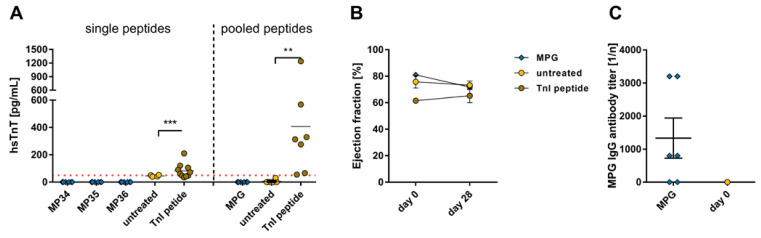
Evaluation of cardiac damage and dysfunction after immunization with SARS-CoV-2 membrane protein peptide and membrane protein peptide pools. A/JOla mice were immunized with 150 µg peptide on days 0, 7, and 14 [MP34-36 n = 6, TnI n = 10, untreated n = 4; MPG n = 6, TnI n = 7, untreated n = 6]. (**A**) Determination of hsTnT-levels as a cardiac damage marker in blood serum of MP and MPG peptide immunized mice collected on day 28 [threshold (dot line): 50 pg/mL]. (**B**) Analysis of cardiac dysfunction via echocardiographic assessment of ejection fraction using M-Mode over a period of 28 days. (**C**) Antibody production directed against MPG peptides on day 0 and after a 28-day period. All values are indicated as mean ± SEM. For statistics, a Mann–Whitney test (**A**,**C**) or ordinary two-way ANOVA (**B**) was performed, and significant values were marked (** *p* < 0.01, *** *p* < 0.001).

**Figure 5 ijms-24-09873-f005:**
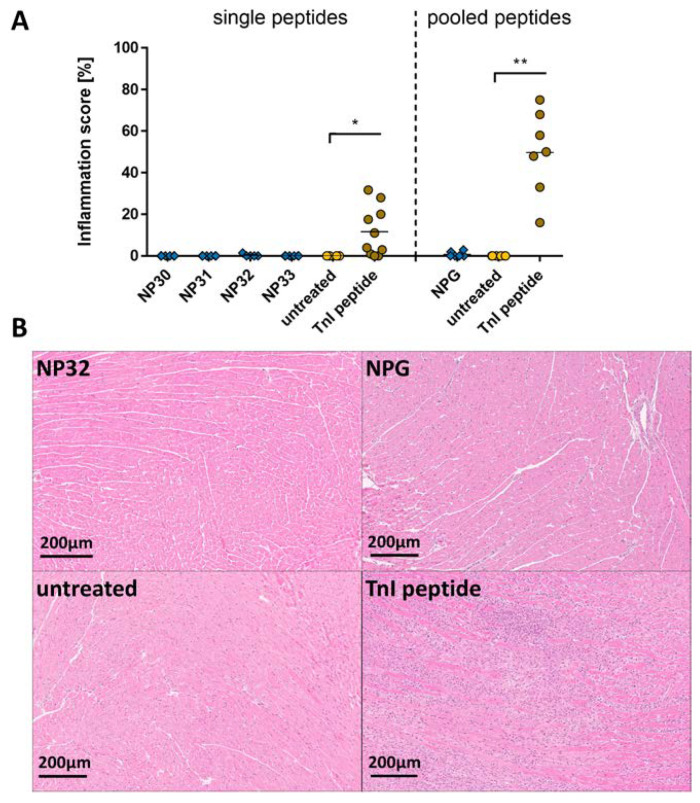
Evaluation of inflammation after immunization with SARS-CoV-2 nucleocapsid protein peptides and peptide pools. A/JOla mice were immunized with 150 µg peptide three times [NP30-33 n = 6, TnI n = 10, untreated n = 4; NPG n = 6, TnI n = 7, untreated n = 6]. (**A**) Inflammation score of single NP peptide (**left**) and pooled NP peptide (**right**) immunized A/JOla mice in percentage based on HE staining scored by two experienced investigators (**B**) Representative images of paraffin-embedded murine heart tissue sections of 3 µm stained with HE at a 20-fold magnification. All values are indicated as mean ± SEM. For statistics, a Mann–Whitney test was performed, and significant values were marked (* *p* < 0.05 and ** *p* < 0.01).

**Figure 6 ijms-24-09873-f006:**
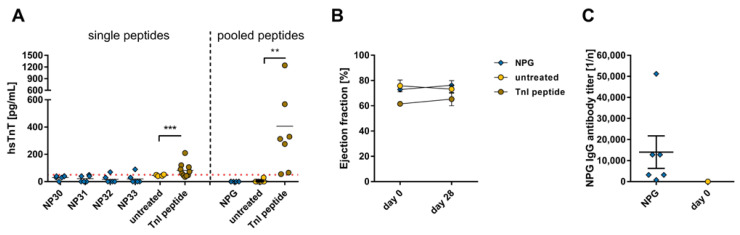
Evaluation of cardiac damage and dysfunction after immunization with SARS-CoV-2 nucleocapsid protein peptide and membrane protein peptide pools. A/JOla mice were immunized with 150 µg peptide on days 0, 7, and 14 [NP30-33 n = 6, TnI n = 10, untreated n = 4; NPG n = 6, TnI n = 7, untreated n = 6]. (**A**) Determination of hsTnT-levels as a cardiac damage marker in blood serum of NP and NPG peptide immunized mice collected on day 28 [threshold (dot line): 50 pg/mL]. (**B**) Analysis of cardiac dysfunction via echocardiographic assessment of ejection fraction using M-Mode over a period of 28 days. (**C**) Antibody production directed against NPG peptides on day 0 and after a 28-day period. All values are indicated as mean ± SEM. For statistics, a Mann–Whitney test (**A**,**C**) or ordinary two-way ANOVA (**B**) was performed, and significant values were marked (** *p* < 0.01, *** *p* < 0.001).

**Figure 7 ijms-24-09873-f007:**
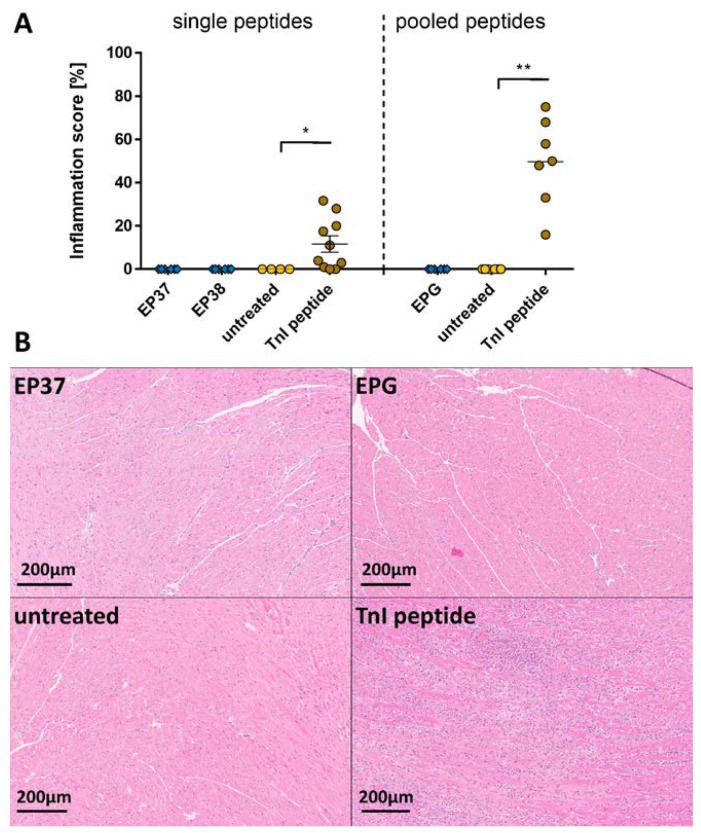
Evaluation of inflammation after immunization with SARS-CoV-2 envelope protein peptides and peptide pools. A/JOla mice were immunized with 150 µg peptide three times [EP37-38n = 6, TnI n = 10, untreated n = 4; EPG n = 6, TnI n = 7, untreated n = 6]. (**A**) Inflammation score of single EP peptide (**left**) and pooled EP peptide (**right**) immunized A/JOla mice in percentage based on HE staining scored by two experienced investigators (**B**) Representative images of paraffin-embedded murine heart tissue sections of 3 µm stained with HE at a 20-fold magnification. All values are indicated as mean ± SEM. For statistics, a Mann–Whitney test was performed, and significant values were marked (* *p* < 0.05 and ** *p* < 0.01).

**Figure 8 ijms-24-09873-f008:**
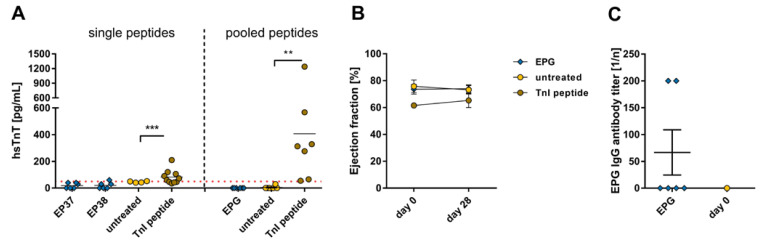
Evaluation of cardiac damage and dysfunction after immunization with SARS-CoV-2 envelope protein peptide and membrane protein peptide pools. A/JOla mice were immunized with 150 µg peptide on days 0, 7, and 14 [EP37-38 n = 6, TnI n = 10, untreated n = 4; EPG n = 6, TnI n = 7, untreated n = 6]. (**A**) Determination of hsTnT-levels as a cardiac damage marker in blood serum of EP and EPG peptide immunized mice collected on day 28 [threshold (dot line): 50 pg/mL]. (**B**) Analysis of cardiac dysfunction via echocardiographic assessment of ejection fraction using M-Mode over a period of 28 days. (**C**) Antibody production directed against EPG peptides on day 0 and after a 28-day period. All values are indicated as mean ± SEM. For statistics, a Mann–Whitney test (**A**,**C**) or ordinary two-way ANOVA (**B**) was performed, and significant values were marked (** *p* < 0.01, *** *p* < 0.001).

**Figure 9 ijms-24-09873-f009:**
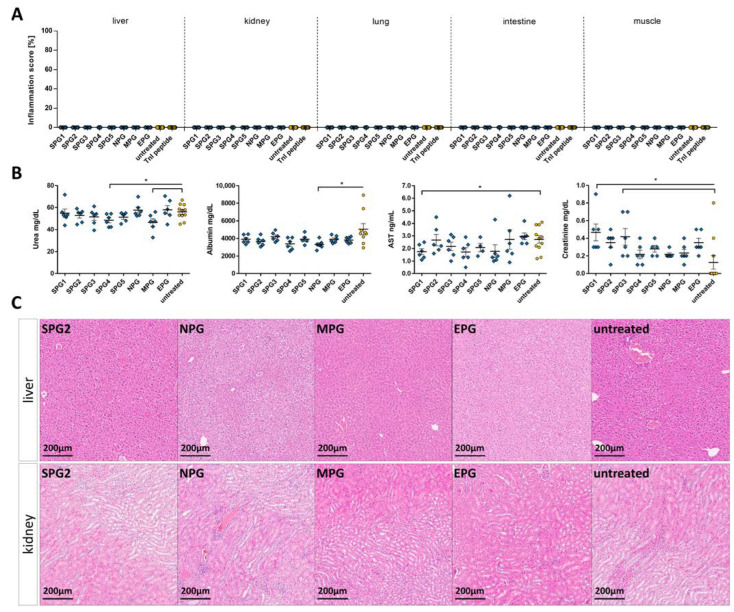
Evaluation of organ integrity and function after immunization with pooled SARS-CoV-2 peptide pools SPG, NGP, MPG, and EPG. (**A**) Inflammation score of liver, kidney, lung, intestine, and muscle of immunized A/JOla mice in percentage based on HE staining scored by two experienced investigators. Therefore, the ratio of the inflamed area to the whole section was determined. (**B**) Investigation of organ function in mice immunized with pooled SARS-CoV-2 peptide sequences. Detection of urea, albumin, creatinine, and AST in blood serum on day 28. (**C**) Histopathological analysis of the liver and kidney in 20× magnification. 3 µm tissue sections were paraffin-embedded and stained with HE. All values are indicated as mean ± SEM. For statistics, a Mann–Whitney test (**A**,**B**) was performed, and significant values were marked (* *p* < 0.05).

**Figure 10 ijms-24-09873-f010:**
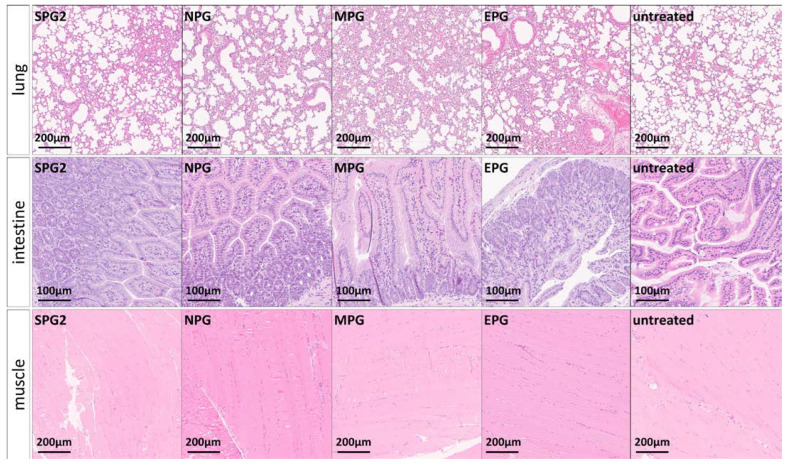
Representative images of lung, intestine, and muscle after immunization with SARS-CoV-2 peptide pools SPG, NPG, MPG, and EPG at 20× magnification. 3 µm tissue sections were paraffin-embedded and stained with HE.

## Data Availability

The raw data supporting the conclusions of this article will be made available by the authors without undue reservation.

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
