# Peer review of "No Evidence for Myocarditis or Other Organ Affection by Induction of an Immune Response against Critical SARS-CoV-2 Protein Epitopes in a Mouse Model Susceptible for Autoimmunity"

_ijms, 2023, doi:10.3390/ijms24129873_

Round 1
Reviewer 1 Report
The study analyzed SARS-CoV-2 epitopes exposure by stimulating A/J mice with different spike, membrane, nucleocapsid and envelope protein peptides and determined the severity of inflammation in different organs. The robustlyscreening of pathological changes in clinical SARS-CoV-2-affected organs were highly appreciated, but here are still some comments.
1. Is the amount of the peptide usage in the entire experiments equivalent the long-term SARS-CoV-2 infection produced peptide concentration? 2. The current study was used three times of stimulation to 14 days, here are some papers suggesting 28 days or 31 days stimulation as long-tern Covid-19 animal models. Can the authors provide the justice of using the treatment to induce long-term of Covid? The concern is if the negative results indeed represent long Covid stimulation. DOI:10.1126/scitranslmed.abq3059
https://doi.org/10.1371/journal.ppat.1010741
DOI: 10.1038/s41587-021-01155-4
3. Please explain the reason why in Figure 1A, 3A, 4A, 5A, 6A, 7A, the pooled experiment and single peptide stimulation, the Tnl stimulation concentration is the identical, but the inflammation score had huge differences. 4. Please do not minimize the differences among groups. For instance, Figure 2 A, Figure 9A, Supplement Figure 6. The Y-axis value is too high to see the fluctuation among groups. Though there were no statistic differences, but readers even could not see exact values in the graph.Author Response
Please see the attachment for point-to-point response to the reviewers.

Reviewer 2 Report
I thank the authors for the interesting results they present in this article.
The methods used in this work are not able to detect autoimmune processes. HE stain provides only a general overview of a tissue sample's structure. The sections show inflammation, but it is not an autoimmune process. You can only talk about inflammation, but not about autoimmune processes.
Peptides are small molecules. Therefore, they are weak immunogens. To increase immunogenicity, peptides should be immobilized on the protein, for example, BSA. It would be interesting to compare the results obtained in this work with the results of immunization of mice with immobilized peptides.
Please use "longCOVID" or "long-COVID"
Line 63: please decipher “persistent symptoms”
Lines 64-66: the meaning of this sentence is unclear.
Lines 82-84: what is “pre-exposure”? Please list “specific autoantibodies”.
Lines 88-90: please add reference to Murphy et al
Paragraph 98-109 is a collection of unrelated facts. Change this paragraph.
Lines 130-131: please add characterization of A/JOla mice. Why are these mice chosen?
What is the composition of the peptide emulsion? Why were these peptides chosen? What is the length of the peptides? Why was this length chosen?
Why was the TnI-peptide chosen as a positive control?
Lines 131-134: it is not clear how many mice were in each group.
Lines 141, 144, 146, 158, 160, 163 et al: reference source not found
How was the percentage of inflammation in the myocardium on sections estimated? Add details.
156: What is hsTnT?
section 2.3 Not clear how organ dysfunction was assessed? Only by using ELISA?
Rice. 2D, 4C, 6C. What is the unit of measurement for antibody titer in blood plasma?
The paper does not present the results of echocardiographic studies of animals, although they are mentioned in the discussion section.
453. Who or what is Envigo?
475 hsTnT analysis method by electrochemiluminescence is not described in sufficient detail
How was the percentage of inflammation in the myocardium on sections estimated? Add details.
156: What is hsTnT?
Section 2.3 Not clear how organ dysfunction was assessed? Only by using ELISA?
Rice 2D, 4C, 6C. What is the unit of measurement for antibody titer in blood plasma?
The paper does not present the results of echocardiographic studies of animals, although they are mentioned in the discussion section.
453. Who or what is Envigo?
475 hsTnT analysis method by electrochemiluminescence is not described in sufficient detail
Author Response
Please see the attachment for point-to-point response to the reviewers.

Reviewer 3 Report
Thanks to the authors for a great and interesting work! These results are promising for combating the consequences of SARS-CoV-2. After minor corrections, the article can be published. Please continue.
Minor mistakes
Lowercase letter “b” (line 84); upper case “B” (line 421); upper case “P” (line 463); extra letter “o” (line 508); extra brackets (line 322), extra letter (‘immounorelevant’, line 123), extra letter “P” (Figure 3, A).
Either separate "1:" from the rest of the digits, or leave it out: lines 168, 214.
It would be better if all units of measurement in the figures were displayed in the same way
Problem with links to tables and figures
Make links to tables and figures in text:
lines 141, 144, 146, 158, 160, 163, 168, 191, 193, 194, 206, 208, 210, 214, 236, 237, 239, 249, 252, 256, 258, 280, 282, 283, 295, 298, 300, 320, 322, 329, 332, 335, 338.
Instead of what is necessary, it is written: “Error! Reference source not found”.
Lines 320: I guess you mean ‘Figure-S1-5’
Abbreviation
Abbreviations must be deciphered at the first mention:
lines 40 (MERS-CoV), 43 (hsTnT, CPR), 49 (ACE-2), 67 (CDC), 121 (EAM), 213 (NPG), 338 (UTR), 339 (CKM).
The name of the mouse strain should be standardized ( e.g. line 130 (A/Jola) and line 149 (A/J))
‘The MPG IgG antibody titer in MPG 212 immunized mice turned out lower compared to the SPG and NPG groups’ (lines 112-123) – this is an abbreviation and this conclusion does not correspond to the sequence of presentation.
Unclear meaning
Perhaps you meant something else, if not, please explain:
‘…is interfered by the docking of SARS-CoV-2’ (line 54) – docking is a method which predicts the preferred orientation of one molecule to a second when a ligand and a target are bound to each other to form a stable complex.
‘…vaccination is also depending on the sex, which differ in their seroreversion’ (line 99) – ‘Seroreversion’ is a change in HIV status from positive to negative. ‘Seroconversion’ is the interval between infection and the appearance of antibodies in the blood.
Figure captions
‘…by two experienced readers’ (lines 152, 201, 245, 290, 343 – maybe it's better to replace the word ‘readers’ with the word ‘investigators’ (as in Materials and Methods)
It is better to indicate which part of the heart was considered – lines 153, 202, 246, 290
It is necessary to decipher the abbreviations in the figure (d0, d28) – lines 178, 223, 268, 310
Somewhere in the text or in each figure captions indicate how you defined threshold (e.g. before Figure 2)
Indicate which parts of the figure these criteria apply to (line 180, e.g. ‘Mann-Whitney (A,C)’)
Figure 9: no data on statistical processing, add.
Figure 9, Figure 10: it is necessary to indicate which parts of the organs were considered; it is desirable to explain somewhere why there is no positive control, and in the description of the figure, explain how the researcher determines the degree of infiltration.
Text perception
‘…once a week for three times’ (line 130) – it would be nice to briefly explain once why mice were immunized in this way or leave a reference to the study.
‘…animals immunized with pooled SP peptides’ (line 143) – it is not clear what pools are. Leave a link here to Table S1.
‘The MPG IgG antibody titer in MPG 212 immunized mice turned out lower compared to the SPG and NPG groups’ (lines 112-123) – is this conclusion statistically supported? Data is required for the conclusion to be clear.
‘Urea plasma levels were scattered and we could detect a significant difference between SPG4 (48.48±2.251 vs. 56.07±2.056, p*=0.0341, 328 t-test) and MPG (46.52±3.339 vs. 56.07±2.056, p*=0.0209, t-test…’ – for a better understanding, place the word ‘(control)’ after the highlighted numbers.
‘Our results showed increased creatinine concentrations in SPG1 mice compared to controls (p*=0.0172, t-test), as well as SPG3 (p*=0.0369, t-test)’ (line 333) – this output does not match the Figure 9 (B).
‘UTR’ (line 338) – what is it? This designation is no longer used in the text.
Explanations of how the concentration of the substances that you consider (albumin, etc.) is related to the state of the organs should be briefly placed in the Introduction or Results (lines 393-400, 403-409).
‘No difference could be found between the sexes, as neither the males nor the females showed signs of inflammation, dysfunction or tissue damage in any of the organs examined’ – this is an interesting conclusion. For some reason you chose not to make it part of the results, but talk about it in the Abstract. Add the data on which you made this conclusion.
It would be nice to point out the limitations of the study in the Discussion.
Materials and methods
It would be great if you provide a plan for your research in order to better understand the logic of the work.
Indicate the sex ratio or show that it is not important (line 467).
‘(1-38)’ (line 463) – it's not clear what it is. Refer to Table S1.
Histopathological Analysis (line 480) – the method is only described for the heart, but you have explored other organs as well. Make a description of the methods for other organs.
‘Groups were analyzed using Mann-Whitney test for parametric data’ (line 530-531) – for this criterion, the normal distribution of samples is not required. Are you sure that's what you meant to write? Reformulate the sentence, it has an unclear message.
‘2-way analysis of variance’ (line 532) – it's better to write an abbreviation.
‘…were subcutaneously immunized three times at 7-day intervals (day 0, 7, 14)…’ (line 468) – explain why you used this timetable in your work. In the paper you referred ([65]) other time lapses were used.
Abstract
‘Furthermore, we investigated sex-specific differences’ (line 23) – in the article you did not provide data for this. It is necessary either to include the study of sex differences in the results, or to remove the phrase from the abstract.
Supplementary
Add the name of the area under consideration to the figure captions.
References
‘SARS-CoV-2 infection can also lead to inflammation of the endothelium, which serves as a protective layer in blood vessels [11]’ (lines 44-45) – Are you sure psychological distress among Chinese people is related to endothelial inflammation?
‘This can lead not only to pneumonia, but also to systemic endothelitis, which subsequently spreads to the heart [12]’ (lines 46-47) – the article on this link does not correspond to the thesis presented.
[22], [34], [76] – references must be provided.
English language is very good, just minor editing may be required.
Author Response

(The authors gave the same response as above.)

Round 2
Reviewer 2 Report
The new additions to the manuscript made a big difference. The quality of the paper had improved, and all my questions were addressed. No more comments.